# Learning Bayesian networks with ancestral constraints

**Eunice Yuh-Jie Chen** and **Yujia Shen** and **Arthur Choi** and **Adnan Darwiche**
Computer Science Department
University of California
Los Angeles, CA 90095
{eyjchen,yujias,aychoi,darwiche}@cs.ucla.edu

## Abstract

We consider the problem of learning Bayesian networks optimally, when subject to background knowledge in the form of ancestral constraints. Our approach is based on a recently proposed framework for optimal structure learning based on non-decomposable scores, which is general enough to accommodate ancestral constraints. The proposed framework exploits oracles for learning structures using decomposable scores, which cannot accommodate ancestral constraints since they are non-decomposable. We show how to empower these oracles by passing them decomposable constraints that they can handle, which are inferred from ancestral constraints that they cannot handle. Empirically, we demonstrate that our approach can be orders-of-magnitude more efficient than alternative frameworks, such as those based on integer linear programming.

## 1 Introduction

Bayesian networks learned from data are broadly used for classification, clustering, feature selection, and to determine associations and dependencies between random variables, in addition to discovering causes and effects; see, e.g., [Darwiche, 2009, Koller and Friedman, 2009, Murphy, 2012].

In this paper, we consider the task of learning Bayesian networks optimally, subject to background knowledge in the form of *ancestral constraints*. Such constraints are important in practice as they allow one to assert direct or indirect cause-and-effect relationships (or lack thereof) between random variables. Further, one expects that their presence should improve the efficiency of the learning process as they reduce the size of the search space. However, nearly all mainstream approaches for optimal structure learning make a fundamental assumption, that the scoring function (i.e., the prior and likelihood) is decomposable. This in turn limits their ability to integrate ancestral constraints, which are non-decomposable. Such approaches only support structure-modular constraints such as the presence or absence of edges, or order-modular constraints such as pairwise constraints on topological orderings; see, e.g., [Koivisto and Sood, 2004, Parviainen and Koivisto, 2013].

Recently, a new framework has been proposed for optimal Bayesian network structure learning [Chen et al., 2015], but with *non-decomposable priors and scores*. This approach is based on navigating the seemingly intractable search space over all network structures (i.e., all DAGs). This intractability can be mitigated however by leveraging an *omniscient oracle* that can optimally learn structures with *decomposable scores*. This approach led to the first system for finding optimal DAGs (i.e., model selection) given order-modular priors (a type of non-decomposable prior) [Chen et al., 2015]. The approach was also applied towards the enumeration of the $k$-best structures [Chen et al., 2015, 2016], where it was orders-of-magnitude more efficient than the existing state-of-the-art [Tian et al., 2010, Cussens et al., 2013, Chen and Tian, 2014].

In this paper, we show how to incorporate non-decomposable constraints into the structure learning approach of Chen et al. [2015, 2016]. We consider learning with ancestral constraints, and inferring decomposable constraints from ancestral constraints to empower the oracle. In principle, structure learning approaches based on integer linear programming (ILP) and constraint programming (CP) can also represent ancestral constraints (and other non-decomposable constraints) [Jaakkola et al., 2010, Bartlett and Cussens, 2015, van Beek and Hoffmann, 2015].[1] We empirically evaluate the proposed approach against those based on ILP, showing *orders of magnitude* improvements.

This paper is organized as follows. In Section 2, we review the problem of Bayesian network structure learning. In Section 3, we discuss ancestral constraints and how they relate to existing structure learning approaches. In Section 4, we introduce our approach for learning with ancestral constraints. In Section 5, we show how to infer decomposable constraints from non-decomposable ancestral constraints. We evaluate our approach empirically in Section 6, and conclude in Section 7.

## 2   Technical preliminaries

We use upper case letters $X$ to denote variables and bold-face upper case letters $\mathbf{X}$ to denote sets of variables. We use $X$ to denote a variable in a Bayesian network and $\mathbf{U}$ to denote its parents.

In score-based approaches to structure learning, we are given a complete dataset $\mathcal{D}$ and want to learn a DAG $G$ that optimizes a *decomposable score*, which aggregates scores over the DAG families $X\mathbf{U}$:

$$\mathsf{score}(G \mid \mathcal{D}) = \sum_{X\mathbf{U}} \mathsf{score}(X\mathbf{U} \mid \mathcal{D}) \tag{1}$$

The MDL and BDeu scores are examples of decomposable scores; see, e.g., Darwiche [2009], Koller and Friedman [2009], Murphy [2012]. The seminal к2 algorithm is one of the first algorithms to exploit decomposable scores [Cooper and Herskovits, 1992]. The к2 algorithm optimizes Equation 1, but assumes that a DAG $G$ is consistent with a given topological ordering $\sigma$. This assumption decomposes the structure learning problem into independent sub-problems, where we find the optimal set of parents for each variable $X$, from those variables that precede $X$ in ordering $\sigma$.

We can find the DAG $G$ that optimizes Equation 1 by running the к2 algorithm on all $n!$ variable orderings $\sigma$, and then take the DAG with the best score. Note that these $n!$ instances share many computational sub-problems: finding the optimal set of parents for some variable $X$. One can aggregate these common sub-problems, leaving us with only $n \cdot 2^{n-1}$ unique sub-problems. This technique underlies a number of modern approaches to score-based structure learning, including some based on dynamic programming [Koivisto and Sood, 2004, Singh and Moore, 2005, Silander and Myllymäki, 2006], and related approaches based on heuristic search methods such as A* [Yuan et al., 2011, Yuan and Malone, 2013]. This aggregation of к2 sub-problems also corresponds to a search space called the *order graph* [Yuan et al., 2011, Yuan and Malone, 2013].

Bayesian network structure learning can also be formulated using integer linear programming (ILP), with Equation 1 as the linear objective function of an ILP. Further, for each variable $X$ and candidate parent set $\mathbf{U}$, we introduce an ILP variable $I(X, \mathbf{U}) \in \{0, 1\}$ to represent the event that $X$ has parents $\mathbf{U}$ when $I(X, \mathbf{U}) = 1$, and $I(X, \mathbf{U}) = 0$ otherwise. We then assert constraints that each variable $X$ has a unique set of parents, $\sum_{\mathbf{U}} I(X, \mathbf{U}) = 1$. Another set of constraints ensure that all variables $X$ and their parents $\mathbf{U}$ must yield an acyclic graph. One approach is to use cluster constraints [Jaakkola et al., 2010], where for each cluster $\mathbf{C} \subseteq \mathbf{X}$, at least one variable $X$ in $\mathbf{C}$ has no parents in $\mathbf{C}$, $\sum_{X \in \mathbf{C}} \sum_{\mathbf{U} \cap \mathbf{C} = \emptyset} I(X, \mathbf{U}) \geq 1$. Finally, we have the objective function of our ILP, $\sum_{X \in \mathbf{X}} \sum_{\mathbf{U} \subseteq \mathbf{X} \setminus X} \mathsf{score}(X\mathbf{U} \mid \mathcal{D}) \cdot I(X, \mathbf{U})$, which corresponds to Equation 1.

## 3   Ancestral constraints

An ancestral constraint specifies a relation between two variables $X$ and $Y$ in a DAG $G$. If $X$ is an ancestor of $Y$, then there is a directed path connecting $X$ to $Y$ in $G$. If $X$ is not an ancestor of $Y$, then there is no such path. Ancestral constraints can be used, for example, to express background knowledge in the form of cause-and-effect relations between variables. When $X$ is an ancestor of $Y$, we have a *positive* ancestral constraint, denoted $X \rightsquigarrow Y$. When $X$ is not an ancestor of $Y$, we have a

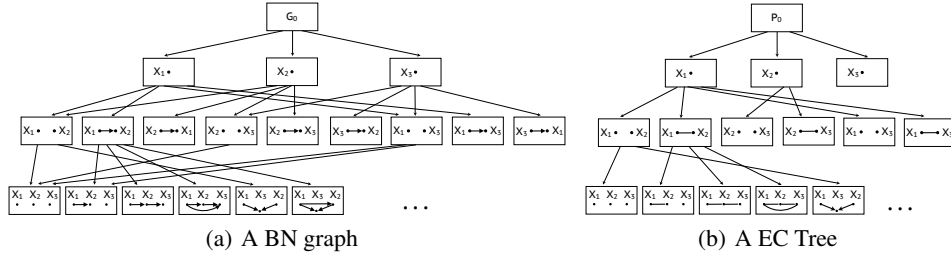

(a) A BN graph                    (b) A EC Tree

Figure 1: Bayesian network search spaces for the set of variables $\mathbf{X} = \{X_1, X_2, X_3\}$.

*negative* ancestral constraint, denoted $X \not\rightsquigarrow Y$. In this case, there is no directed path from $X$ to $Y$, but there may still be a directed path from $Y$ to $X$. Positive ancestral constraints are transitive, i.e., if $X \rightsquigarrow Y$ and $Y \rightsquigarrow Z$ then $X \rightsquigarrow Z$. Negative ancestral constraints are not transitive.

Ancestral constraints are *non-decomposable* since we cannot in general check whether an ancestral constraint is satisfied or violated by *independently* checking the parents of each variable. For example, consider an optimal DAG, compatible with ordering $\langle X_1, X_2, X_3 \rangle$, from the family scores:

| $X$ | $\mathbf{U}$ | score | $X$ | $\mathbf{U}$ | score | $X$ | $\mathbf{U}$ | score |
|---|---|---|---|---|---|---|---|---|
| $X_1$ | $\{\}$ | 1 | $X_2$ | $\{\}, \{X_1\}$ | 1, 2 | $X_3$ | $\{\}, \{X_1\}, \{X_2\}, \{X_1, X_2\}$ | 10, 10, 1, 10 |

The optimal DAG (with minimal score) in this case is $\boxed{X_1 \qquad X_2 \rightarrow X_3}$. If we assert the ancestral constraint $X_1 \rightsquigarrow X_3$, then the optimal DAG is $\boxed{X_1 \rightarrow X_2 \rightarrow X_3}$. Yet, we cannot enforce this ancestral constraints using independent, local constraints on the parents that each variable can take. In particular, the choice of parents for variable $X_2$ and the choice of parents for variable $X_3$ will jointly determine whether $X_1$ is an ancestor of $X_3$. Hence, the K2 algorithm and approaches based on the order graph (dynamic programming and heuristic search) cannot enforce ancestral constraints.

These approaches, however, can enforce *decomposable constraints,* such as the presence or absence of an edge $U \rightarrow X$, or a limit on the size of a family $X\mathbf{U}$. Interestingly, one can infer some decomposable constraints from non-decomposable ones. We discuss this technique extensively later, showing how it can lead to significant impact on the efficiency of structure search.

Structure learning approaches based on ILP can in principle enforce non-decomposable constraints, when they can be encoded as linear constraints. In fact, ancestral relations have been employed in ILPs and other formalisms to enforce a graph's acyclicity; see, e.g., [Cussens, 2008]. However, to our knowledge, these approaches have not been evaluated for learning structures with ancestral constraints. We provide such an empirical evaluation in Section 6.[2]

## 4 Learning with constraints

In this section, we review two recently proposed search spaces for learning Bayesian networks: the BN graph and the EC tree [Chen et al., 2015, 2016]. We subsequently show how we can adapt the EC tree to facilitate the learning of Bayesian network structures under ancestral constraints.

### 4.1 BN graphs

The *BN graph* is a search space for learning structures with *non-decomposable* scores [Chen et al., 2015]. Figure 1(a) shows a BN graph over 3 variables, where nodes represent DAGs over different subsets of variables. A directed edge $G_i \xrightarrow{X\mathbf{U}} G_j$ from a DAG $G_i$ to a DAG $G_j$ exists in the BN graph iff $G_j$ can be obtained from $G_i$ by adding a leaf node $X$ with parents $\mathbf{U}$. Each edge has a cost, corresponding to the score of the family $X\mathbf{U}$, as in Equation 1. Hence, a path from the root $G_0$ to a DAG $G_n$ yields the score of the DAG, $\mathsf{score}(G_n \mid \mathcal{D})$. As a result, the shortest path in the BN graph (the one with the lowest score) corresponds to an optimal DAG, as in Equation 1.

Unlike the order graph, the BN graph explicitly represents all possible DAGs. Hence, ancestral constraints can be easily integrated by *pruning* the search space, i.e., by pruning away those DAGs that do not satisfy the given constraints. Consider Figure 1(a) and the ancestral constraint $X_1 \rightsquigarrow X_2$. Since the DAG $\boxed{X_1 \quad X_2}$ violates the constraint, we can prune this node, along with all of its descendants, as the descendants must also violate an ancestral constraint (adding new leaves to a DAG will not undo a violated ancestral constraint). Finding a shortest path in this pruned search space will yield an optimal Bayesian network satisfying a given set of ancestral constraints.

We can use A* search to find a shortest path in a BN graph. A* is a best-first search algorithm that uses an evaluation function $f$ to guide the search. For a given DAG $G$, we have the evaluation function $f(G) = g(G) + h(G)$, where $g(G)$ is the *actual* cost to reach $G$ from the root $G_0$, and $h(G)$ is the *estimated* cost to reach a leaf from $G$. A* search is guaranteed to find a shortest path when the heuristic function $h$ is *admissible*, i.e., it does not over-estimate. Chen et al. [2015, 2016] showed that a heuristic function can be induced by any learning algorithm that takes a (partial) DAG as input, and returns an *optimal* DAG that extends it. Learning systems based on the order graph fall in this category and can be viewed as *powerful oracles* that help us to navigate the DAG graph. We employed URLEARNING as an oracle in our experiments [Yuan and Malone, 2013]. We will later show how to *empower* this oracle by passing it decomposable constraints that we infer from a set of non-decomposable ancestral constraints—the impact of this empowerment turns out to be dramatic.

## 4.2 EC trees

The *EC tree* is a recently proposed search space that improves the BN graph along two dimensions [Chen et al., 2016]. First, it merges Markov-equivalent nodes in the BN graph. Second, it canonizes the resulting EC graph into a tree, where each node is reachable by a unique path from the root. Two network structures are Markov equivalent iff they have the same undirected skeleton and the same $v$-structures. A Markov equivalence class can be represented by a *completed, partially directed acyclic graph* (CPDAG). The set of structures represented by a CPDAG $P$ is denoted by $\mathsf{class}(P)$ and may contain exponentially many Markov equivalent structures.

Figure 1(b) illustrates an EC tree over 3 variables, where nodes represent CPDAGs over different subsets of variables. A directed edge $P_i \xrightarrow{X\mathbf{U}} P_j$ from a CPDAG $P_i$ to a CPDAG $P_j$ exists in the EC tree iff there exists a DAG $G_j \in \mathsf{class}(P_j)$ that can be obtained from a DAG $G_i \in \mathsf{class}(P_i)$ by adding a leaf node $X$ with parents $\mathbf{U}$, but where $X$ must be the largest variable in $G_j$ (according to some canonical ordering). Each edge of an EC tree has a cost $\mathsf{score}(X\mathbf{U} \mid \mathcal{D})$, so the shortest path in the EC tree corresponds to an optimal equivalence class of Bayesian networks.

## 4.3 EC trees and ancestral constraints

A DAG $G$ satisfies a set of ancestral constraints $\mathcal{A}$ (both over the same set of variables) iff the DAG $G$ satisfies each constraint in $\mathcal{A}$. Moreover, a CPDAG $P$ satisfies $\mathcal{A}$ iff there exists a DAG $G \in \mathsf{class}(P)$ that satisfies $\mathcal{A}$. We enforce ancestral constraints by pruning a CPDAG node $P$ from an EC tree when $P$ does not satisfy the constraints $\mathcal{A}$. First, consider an ancestral constraint $X_1 \not\rightsquigarrow X_2$. A CPDAG $P$ containing a directed path from $X_1$ to $X_2$ violates the constraint, as every structure in $\mathsf{class}(P)$ contains a path from $X_1$ to $X_2$. Next, consider an ancestral constraint $X_1 \rightsquigarrow X_2$. A CPDAG $P$ with no partially directed paths from $X_1$ to $X_2$ violates the given constraint, as no structure in $\mathsf{class}(P)$ contains a path from $X_1$ to $X_2$.[3] Given a CPDAG $P$, we first test for these two cases, which can be done efficiently. If these tests are inconclusive, we exhaustively enumerate the structures of $\mathsf{class}(P)$, to check if any of them satisfies the given constraints. If not, we can prune $P$ and its descendants from the EC tree. The soundness of this pruning step is due to the following.

**Theorem 1** *In an EC tree, a CPDAG $P$ satisfies ancestral constraints $\mathcal{A}$, both over the same set of variables $\mathbf{X}$, iff its descendants satisfy $\mathcal{A}$.*

# 5 Projecting constraints

In this section, we show how one can project *non-decomposable* ancestral constraints onto *decomposable* edge and ordering constraints. For example, if $\mathcal{G}$ is a set of DAGs satisfying a set of ancestral

constraints $\mathcal{A}$, we want to find the edges that appear in all DAGs of $\mathcal{G}$. These *projected constraints* can be then used to improve the efficiency of structure learning. Recall (from Section 4.1) that our approach to structure learning uses a heuristic function that utilizes an optimal structure learning algorithm for decomposable scores (the oracle). We *tighten* this heuristic (empower the oracle) by passing to it projected edge and ordering constraints, leading to a more efficient search when we are subject to non-decomposable ancestral constraints.

Given a set of ancestral constraints $\mathcal{A}$, we shall show how to infer new edge and ordering constraints, that we can utilize to empower our oracle. For the case of edge constraints, we propose a simple algorithm that can efficiently enumerate all inferrable edge constraints. For the case of ordering constraints, we propose a reduction to MaxSAT, that can find a maximally large set of ordering constraints that can be jointly inferred from ancestral constraints.

## 5.1 Edge constraints

We now propose an algorithm for finding all edge constraints that can be inferred from a set of ancestral constraints $\mathcal{A}$. We consider (decomposable) constraints on the presence of an edge, or the absence of an edge. We refer to edge presence constraints as *positive* constraints, denoted by $X \to Y$, and refer to edge absence constraints as *negative* constraints, denoted by $X \not\to Y$.

We let $\mathcal{E}$ denote a set of edge constraints. We further let $\mathcal{G}(\mathcal{A})$ denote the set of DAGs $G$ over the variables $\mathbf{X}$ that satisfy all ancestral constraints in the set $\mathcal{A}$, and let $\mathcal{G}(\mathcal{E})$ denote the set of DAGs $G$ that satisfy all edge constraints in $\mathcal{E}$. Given a set of ancestral constraints $\mathcal{A}$, we say that $\mathcal{A}$ entails a positive edge constraint $X \to Y$ iff $\mathcal{G}(\mathcal{A}) \subseteq \mathcal{G}(X \to Y)$, and that $\mathcal{A}$ entails a negative edge constraint $X \not\to Y$ iff $\mathcal{G}(\mathcal{A}) \subseteq \mathcal{G}(X \not\to Y)$. For example, consider the four DAGs over the variables $X, Y$ and $Z$ that satisfy ancestral constraints $X \not\rightsquigarrow Z$ and $Y \rightsquigarrow Z$.

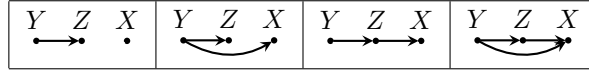

First, we note that no DAG above contains the edge $X \to Z$, since this would immediately violate the constraint $X \not\rightsquigarrow Z$. Next, no DAG above contains the edge $X \to Y$. Suppose instead that this edge appeared; since $Y \rightsquigarrow Z$, we can infer $X \rightsquigarrow Z$, which contradicts the existing constraint $X \not\rightsquigarrow Z$. Hence, we can infer the negative edge constraint $X \not\to Y$. Finally, no DAG above contains the edge $Z \to Y$, since this would lead to a directed cycle with the constraint $Y \rightsquigarrow Z$.

Before we present our algorithm for inferring edge constraints, we first revisit some properties of ancestral constraints that we will need. Note that given a set of ancestral constraints, we may be able to infer additional ancestral constraints. First, given two constraints $X \rightsquigarrow Y$ and $Y \rightsquigarrow Z$, we can infer an additional ancestral constraint $X \rightsquigarrow Z$ (by transitivity of ancestral relations). Second, if adding a path $X \rightsquigarrow Y$ would create a directed cycle (e.g., if $Y \rightsquigarrow X$ exists in $\mathcal{A}$), or if it would violate an existing negative ancestral constraints (e.g., if $X \not\rightsquigarrow Z$ and $Y \rightsquigarrow Z$ exists in $\mathcal{A}$), then we can infer a new negative constraint $X \not\rightsquigarrow Y$. By using a few rules based on the examples above, we can efficiently enumerate all of the ancestral constraints that are entailed by a given set of ancestral constraints (details omitted for space). Hence, we shall subsequently assume that a given set of ancestral constraints $\mathcal{A}$ will already include all ancestral constraints that can be entailed from it. We then refer to $\mathcal{A}$ as a maximum set of ancestral constraints.

We now consider how to infer edge constraints from a (maximum) set of ancestral constraints $\mathcal{A}$. First, let $\alpha(X)$ be the set that consists of $X$ and every $X'$ such that $X' \rightsquigarrow X \in \mathcal{A}$, and let $\beta(X)$ be the set that consists of $X$ and every $X'$ such that $X \rightsquigarrow X' \in \mathcal{A}$. In other words, $\alpha(X)$ contains $X$ and all nodes that are constrained to be ancestors of $X$ by $\mathcal{A}$, i.e., each $X' \in \alpha(X)$ is either $X$ or an ancestor of $X$, for all DAGs $G \in \mathcal{G}(\mathcal{A})$. Similarly, $\beta(X)$ contains $X$ and all nodes that are constrained to be descendants of $X$ by $\mathcal{A}$.

First, we can check if a negative edge constraint $X \not\to Y$ is entailed by $\mathcal{A}$ by enumerating all possible $X_a \not\rightsquigarrow Y_b$ for all $X_a \in \alpha(X)$ and all $Y_b \in \beta(Y)$. If any $X_a \not\rightsquigarrow Y_b$ is in $\mathcal{A}$ then we know that $\mathcal{A}$ entails $X \not\to Y$. That is, since $X_a \rightsquigarrow X$ and $Y \rightsquigarrow Y_b$, then if there was a DAG $G \in \mathcal{G}(\mathcal{A})$ with the edge $X \to Y$, then $G$ would also have a path from $X_a$ to $Y_b$. Hence, we can infer $X \not\to Y$. This idea is summarized by the following theorem:

**Theorem 2** *Given a maximum set of ancestral constraints $\mathcal{A}$, then $\mathcal{A}$ entails the negative edge constraint $X \nrightarrow Y$ iff $X_a \nrightarrow Y_b$, where $X_a \in \alpha(X)$ and $Y_b \in \beta(Y)$.*

Next, suppose that both (1) $\mathcal{A}$ dictates that $X$ can reach $Y$, and that (2) $\mathcal{A}$ dictates that there is no path from $X$ to $Z$ to $Y$, for any other variable $Z$. In this case, we can infer a positive edge constraint $X \rightarrow Y$. We can again verify if $X \rightarrow Y$ is entailed by $\mathcal{A}$ by enumerating all relevant candidates $Z$, based on the following theorem.

**Theorem 3** *Given a maximum set of ancestral constraints $\mathcal{A}$, then $\mathcal{A}$ entails the positive edge constraint $X \rightarrow Y$ iff $\mathcal{A}$ contains $X \rightsquigarrow Y$ and for all $Z \notin \alpha(X) \cup \beta(Y)$, the set $\mathcal{A}$ contains a constraint $X_a \nrightarrow Z_b$ or $Z_a \nrightarrow Y_b$, where $X_a \in \alpha(X), Z_b \in \beta(Z), Z_a \in \alpha(Z)$ and $Y_b \in \beta(Y)$.*

## 5.2 Topological ordering constraints

We next consider constraints on the topological orderings of a DAG. An ordering satisfies a constraint $X < Y$ iff $X$ appears before $Y$ in the ordering. Further, an ordering constraint $X < Y$ is *compatible* with a DAG $G$ iff there exists a topological ordering of DAG $G$ that satisfies the constraint $X < Y$. The negation of an ordering constraint $X < Y$ is the ordering constraint $Y < X$. A given ordering satisfies either $X < Y$ or $Y < X$, but not both at the same time. A DAG $G$ may be compatible with both $X < Y$ and $Y < X$ through two different topological orderings.

We let $\mathcal{O}$ denote a set of ordering constraints, and let $\mathcal{G}(\mathcal{O})$ denote the set of DAGs $G$ that are compatible with each ordering constraint in $\mathcal{O}$. The task of determining whether a set of ordering constraints $\mathcal{O}$ is entailed by a set of ancestral constraints $\mathcal{A}$, i.e., whether $\mathcal{G}(\mathcal{A}) \subseteq \mathcal{G}(\mathcal{O})$, is more subtle than the case of edge constraints. For example, consider the set of ancestral constraints $\mathcal{A} = \{Z \nrightarrow Y, X \nrightarrow Z\}$. We can infer the ordering constraint $Y < Z$ from the first constraint $Z \nrightarrow Y$, and $Z < X$ from the second constraint $X \nrightarrow Z$.[4] If we were to assume both ordering constraints, we could infer the third ordering constraint $Y < X$, by transitivity. However, consider the following DAG $G$ which satisfies $\mathcal{A}$: $\boxed{X \rightarrow Y \quad Z}$. This DAG is compatible with the constraint $Y < Z$ as well as the constraint $Z < X$, but it is not compatible with the constraint $Y < X$. Consider the three topological orderings of the DAG $G$: $\langle X, Y, Z \rangle, \langle X, Z, Y \rangle$ and $\langle Z, X, Y \rangle$. We see that none of the orderings satisfy both ordering constraints at the same time. Hence, if we assume both ordering constraints at the same time, it eliminates all topological orderings of the DAG $G$, and hence the DAG itself. Consider another example over variables $W, X, Y$ and $Z$ with a set of ancestral constraints $\mathcal{A} = \{W \nrightarrow Z, Y \nrightarrow X\}$. The following DAG $G$ satisfies $\mathcal{A}$: $\boxed{W \rightarrow X \quad Y \rightarrow Z}$. However, inferring the ordering constraints $Z < W$ and $X < Y$ from each ancestral constraint of $\mathcal{A}$ leads to a cycle in the above DAG ($W < X < Y < Z < W$), hence eliminating the DAG.

Hence, for a given set of ancestral constraints $\mathcal{A}$, we want to infer from it a set $\mathcal{O}$ of ordering constraints that is as large as possible, but without eliminating any DAGs satisfying $\mathcal{A}$. Roughly, this involves inferring ordering constraints $X < Y$ from ancestral constraints $Y \nrightarrow X$, as long as the ordering constraints do not induce a cycle. We propose to encode the problem as an instance of MaxSAT [Li and Manyà, 2009]. Given a maximum set of ancestral constraints $\mathcal{A}$, we construct a MaxSAT instance where propositional variables represent ordering constraints and ancestral constraints (true if the constraint is present, and false otherwise). The clauses encode the ancestral constraints, as well as constraints to ensure acyclicity. By maximizing the set of satisfied clauses, we then maximize the set of constraints $X < Y$ selected. In turn, the (decomposable) ordering constraints can be to empower an oracle during structure search. Our MaxSAT problem includes hard constraints (1-3), as well as soft constraints (4):

1. **transitivity of orderings**: for all $X < Y, Y < Z$: $(X < Y) \wedge (Y < Z) \Rightarrow (X < Z)$

2. **a necessary condition for orderings:** for all $X < Y$: $(X < Y) \Rightarrow (Y \nrightarrow X)$

3. **a sufficient condition for acyclicity:** for all $X < Y$ and $Z < W$: $(X < Y) \wedge (Z < W) \Rightarrow (X \rightsquigarrow Y) \vee (Z \rightsquigarrow W) \vee (X \rightsquigarrow Z) \vee (Y \rightsquigarrow W) \vee (X \rightsquigarrow W) \vee (Y \nrightarrow Z)$

4. **infer orderings from ancestral constraints:** for all $X \nrightarrow Y$ in $\mathcal{A}$: $(X \nrightarrow Y) \Rightarrow (Y < X)$

| N | n = 10 | | | | | | n = 12 | | | | | | n = 14 | | | | | |
|---|---|---|---|---|---|---|---|---|---|---|---|---|---|---|---|---|---|---|
| | 512 | | 2048 | | 8192 | | 512 | | 2048 | | 8192 | | 512 | | 2048 | | 8192 | |
| $p$ | EC | GOB | EC | GOB | EC | GOB | EC | GOB | EC | GOB | EC | GOB | EC | GOB | EC | GOB | EC | GOB |
| 0.00 | < | 7.81 | < | 112.98 | < | 19.70 | 0.01 | 70.85 | 0.02 | 98.28 | 0.02 | 144.21 | 0.06 | 625.081 | 0.07 | 839.46 | 0.09 | 1349.24 |
| 0.01 | < | 9.61 | < | 15.41 | < | 23.58 | 0.01 | 73.39 | 0.01 | 99.46 | 0.01 | 145.75 | 0.05 | 673.003 | 0.06 | 901.50 | 0.08 | 1356.63 |
| 0.05 | < | 11.56 | < | 14.54 | < | 19.85 | 0.02 | 60.16 | 0.01 | 75.40 | 0.27 | 95.11 | 0.08 | 243.681 | 0.05 | 287.45 | 0.04 | 411.22 |
| 0.10 | < | 10.74 | < | 11.60 | < | 13.87 | 0.21 | 52.02 | 0.10 | 53.29 | 0.36 | 59.42 | 0.58 | 176.500 | 1.26 | 198.18 | 0.03 | 218.94 |
| 0.25 | 0.01 | 4.04 | < | 3.43 | < | 3.37 | 4.91 | 22.47 | 0.18 | 20.88 | 0.17 | 19.68 | 55.07 | 126.312 | 0.91 | 112.80 | 0.02 | 107.44 |
| 0.50 | < | 0.87 | < | 0.71 | < | 0.72 | 0.51 | 6.11 | 0.03 | 6.10 | 0.01 | 5.85 | 0.48 | 73.236 | 0.02 | 67.29 | < | 62.60 |
| 0.75 | < | 0.31 | < | 0.75 | < | 0.30 | < | 2.66 | < | 2.62 | < | 2.57 | < | 44.074 | < | 42.95 | < | 41.21 |
| 1.00 | < | 0.21 | < | 0.31 | < | 0.21 | < | 2.29 | < | 2.30 | < | 2.27 | < | 39.484 | < | 39.67 | < | 37.78 |

Table 1: Time (in sec) used by EC tree and GOBNILP to find optimal networks. < is less than 0.01 sec. $n$ is the variable number, $N$ is the dataset size, $p$ is the percentage of the ancestral constraints.

| N | n = 12 | | | | | | | | | n = 14 | | | | | | | | |
|---|---|---|---|---|---|---|---|---|---|---|---|---|---|---|---|---|---|---|
| | 512 | | | 2048 | | | 8192 | | | 512 | | | 2048 | | | 8192 | | |
| $p$ | EC | (t/s) | GOB | EC | (t/s) | GOB | EC | (t/s) | GOB | EC | (t/s) | GOB | EC | (t/s) | GOB | EC | (t/s) | GOB |
| 0.01 | 0.01 | 1 | 63.53 | 0.01 | 1 | 83.59 | 0.02 | 1 | 128.23 | 0.01 | 1 | 634.19 | 0.123 | 1 | 738.25 | 0.12 | 1 | 1295.90 |
| 0.05 | 0.06 | 1 | 55.20 | 0.03 | 1 | 70.20 | 1.18 | 1 | 90.59 | 0.06 | 1 | 228.57 | 0.868 | 1 | 276.68 | 0.18 | 1 | 404.35 |
| 0.10 | 2.56 | 1 | 50.33 | 2.36 | 1 | 52.80 | 0.91 | 1 | 57.66 | 2.54 | 1 | 174.70 | 34.979 | 0.98 | 183.93 | 0.60 | 1 | 210.12 |
| 0.25 | 70.19 | 0.98 | 23.29 | 4.57 | 1 | 20.74 | 1.63 | 1 | 21.16 | 280.59 | 0.84 | 137.67 | 88.80 | 1 | 126.80 | 1.85 | 1 | 126.24 |
| 0.50 | 137.31 | 1 | 7.74 | 15.53 | 1 | 7.80 | 1.43 | 1 | 7.36 | 609.18 | 0.88 | 90.92 | 35.62 | 1 | 85.58 | 4.74 | 1 | 83.81 |
| 0.75 | 21.86 | 1 | 4.38 | 1.73 | 1 | 4.39 | 0.50 | 1 | 4.30 | 258.80 | 1 | 64.51 | 6.49 | 1 | 63.68 | 2.28 | 1 | 64.04 |
| 1.00 | 2.31 | 1 | 4.10 | 0.35 | 1 | 4.07 | 0.15 | 1 | 4.02 | 21.18 | 1 | 61.44 | 1.39 | 1 | 60.56 | 0.54 | 1 | 61.06 |

Table 2: Time $t$ (in sec) used by EC tree and GOBNILP to find optimal networks, without any projected constraints, using a 32G memory and 2 hour time limit. $s$ is the percentage of test cases that finish.

We remark that the above constraints are sufficient for finding a set of ordering constraints $\mathcal{O}$ that are entailed by a set of ancestral constraints $\mathcal{A}$, which is formalized in the following theorem.

**Theorem 4** *Given a maximum set of ancestral constraints $\mathcal{A}$, and let $\mathcal{O}$ be a closed set of ordering constraints. The set $\mathcal{O}$ is entailed by $\mathcal{A}$ if $\mathcal{O}$ satisfies the following two statements:*

1. *for all $X < Y$ in $\mathcal{O}$, $\mathcal{A}$ contains $Y \not\rightsquigarrow X$*

2. *for all $X < Y$ and $Z < W$ in $\mathcal{O}$, where $X, Y, Z$ and $W$ are distinct, $\mathcal{A}$ contains at least one of $X \rightsquigarrow Y, Z \rightsquigarrow W, X \rightsquigarrow Z, Y \rightsquigarrow W, X \rightsquigarrow W, Y \not\rightsquigarrow Z$.*

## 6 Experiments

We now empirically evaluate the effectiveness of our approach to learning with ancestral constraints. We simulated different structure learning problems from standard Bayesian network benchmarks[5] ALARM, ANDES, CHILD, CPCS54, and HEPAR2, by (1) taking a random sub-network $\mathcal{N}$ of a given size[6] (2) simulating a training dataset from $\mathcal{N}$ of varying sizes (3) simulating a set of ancestral constraints of a given size, by randomly selecting ordered pairs whose ground-truth ancestral relations in $\mathcal{N}$ were used as constraints. In our experiments, we varied the number of variables in the learning problem ($n$), the size of the training dataset ($N$), and the percentage of the $n(n-1)/2$ total ancestral relations that were given as constraints ($p$). We report results that were averaged over 50 different datasets: 5 datasets were simulated from each of 2 different sub-networks, which were taken from each of the 5 original networks mentioned above. Our experiments were run on a 2.67GHz Intel Xeon X5650 CPU. We assumed BDeu scores with an equivalent sample size of 1. We further pre-computed the scores of candidate parent sets, which were fed as input into each system evaluated. Finally, we used the EVASOLVER partial MaxSAT solver, for inferring ordering constraints.[7]

In our first set of experiments, we compared our approach with the ILP-based system of GOBNILP,[8] where we encoded ancestral constraints using linear constraints, based on [Cussens, 2008]; note again that both are exact approaches for structure learning. In Table 1, we supplied both systems with decomposable constraints inferred via projection (which empowers the oracle for searching the EC tree, and provides redundant constraints for the ILP). In Table 2, we withheld the projected

| | n = 18 | | | | | | | | | n = 20 | | | | | | | | |
|---|---|---|---|---|---|---|---|---|---|---|---|---|---|---|---|---|---|---|
| N | 512 | | | 2048 | | | 8192 | | | 512 | | | 2048 | | | 8192 | | |
| p | t | s | Δ | t | s | Δ | t | s | Δ | t | s | Δ | t | s | Δ | t | s | Δ |
| 0.00 | 2.25 | 1 | 16.74 | 2.78 | 1 | 8.32 | 3.11 | 1 | 7.06 | 19.40 | 1 | 23.44 | 20.62 | 1 | 10.60 | 28.22 | 1 | 7.22 |
| 0.01 | 2.22 | 1 | 16.58 | 3.46 | 1 | 8.60 | 3.63 | 1 | 7.38 | 30.38 | 1 | 23.67 | 30.46 | 1 | 10.53 | 34.34 | 1 | 7.09 |
| 0.05 | 41.15 | 0.96 | 15.02 | 2.91 | 0.98 | 6.96 | 2.12 | 1 | 5.56 | 87.74 | 0.96 | 18.44 | 39.25 | 1 | 8.20 | 17.40 | 1 | 5.00 |
| 0.10 | 149.40 | 0.94 | 12.72 | 73.03 | 0.96 | 5.81 | 7.35 | 1 | 3.78 | 492.59 | 0.82 | 14.67 | 185.82 | 0.94 | 7.21 | 24.46 | 0.98 | 3.94 |
| 0.25 | 251.74 | 0.78 | 6.33 | 338.10 | 0.94 | 3.79 | 30.90 | 0.96 | 1.96 | 507.02 | 0.58 | 6.17 | 572.68 | 0.88 | 4.46 | 153.81 | 0.96 | 2.28 |
| 0.50 | 95.18 | 0.98 | 5.49 | 13.92 | 0.98 | 2.69 | 116.29 | 0.98 | 1.24 | 163.19 | 0.88 | 6.36 | 46.43 | 0.96 | 2.19 | 70.15 | 1 | 1.07 |
| 0.75 | 9.07 | 1 | 3.30 | 5.83 | 1 | 1.66 | 0.72 | 1 | 0.72 | 1.47 | 1 | 4.49 | 0.28 | 1 | 1.36 | 0.38 | 1 | 0.60 |
| 1.00 | < | 1 | 0.72 | < | 1 | 0.48 | < | 1 | 0.26 | < | 1 | 2.02 | < | 1 | 0.47 | < | 1 | 0.18 |

Table 3: Time $t$ (in sec) used by EC tree to find optimal networks, with a 32G memory, a 2 hour time limit. $<$ is less than 0.01 sec. $n$ is the variable number, $N$ is the dataset size, $p$ is the percentage of the ancestor constraints, $s$ is the percentage of test cases that finish, $\Delta$ is the edge difference of the learned and true networks.

constraints. In Table 1, our approach is consistently orders-of-magnitude faster than GOBNILP, for almost all values of $n$, $N$ and $p$ that we varied. This difference increased with the number of variables $n$.[9] When we compare Table 2 to Table 1, we see that for the EC tree, the projection of constraints has a significant impact on the efficiency of learning (often by several orders of magnitude). For ILP, there is some mild overhead with a smaller number of variables ($n = 12$), but with a larger number of variables ($n = 14$), there were consistent improvements when projected constraints are used.

Next, we evaluate (1) how introducing ancestral constraints effects the efficiency of search, and (2) how scalable our approach is as we increase the number of variables in the learning problem. In Table 3, we report results where we varied the number of variables $n \in \{16, 18, 20\}$, and asserted a 2 hour time limit and a 32GB memory limit. First, we observe an easy-hard-easy trend as we increase the proportion $p$ of ancestral constraints. When $p$ is small, the learning problem is close to the unconstrained problem, and our oracle serves as an accurate heuristic. When $p$ is large, the problem is highly constrained, and the search space is significantly reduced. In contrast, the ILP approach more consistently became easier as more constraints were provided (from Table 1). As expected, the learning problem becomes more challenging when we increase the number of variables $n$, and when less training data is available. We note that our approach scales to $n = 20$ variables here, which is comparable to the scalability of modern score-based approaches reported in the literature (for BDeu scores); e.g., Yuan and Malone [2013] reported results up to 26 variables (for BDeu scores).

Table 3 also reports the average structural Hamming distance $\Delta$ between the learned network and the ground-truth network used to generate the data. We see that as the dataset size $N$ and the proportion $p$ of constraints available increases, the more accurate the learned model becomes.[10] We remark that a relatively small number of ancestral constraints (say $10\%$–$25\%$) can have a similar impact on the quality of the observed network (relative to the ground-truth), as increasing the amount of data available from 512 to 2048, or from 2048 to 8192. This highlights the impact that background knowledge can have, in contrast to collecting more (potentially expensive) training data.

## 7 Conclusion

We proposed an approach for learning the structure of Bayesian networks optimally, subject to ancestral constraints. These constraints are non-decomposable, posing a particular difficulty for learning approaches for decomposable scores. We utilized a search space for structure learning with non-decomposable scores, called the EC tree, and employ an oracle that optimizes decomposable scores. We proposed a sound and complete method for pruning the EC tree, based on ancestral constraints. We also showed how the employed oracle can be empowered by passing it decomposable constraints inferred from the non-decomposable ancestral constraints. Empirically, we showed that our approach is orders-of-magnitude more efficient compared to learning systems based on ILP.

**Acknowledgments**

This work was partially supported by NSF grant #IIS-1514253 and ONR grant #N00014-15-1-2339.

## Footnotes

[1]To our knowledge, however, the ILP and CP approaches have not been previously evaluated, in terms of their efficacy in structure learning with ancestral constraints.

[2]We also make note of Borboudakis and Tsamardinos [2012], which uses ancestral constraints (path constraints) for *constraint-based* learning methods, such as the PC algorithm. Borboudakis and Tsamardinos [2013] further proposes a prior based on path beliefs (soft constraints), and evaluated using greedy local search.

[3]A partially directed path from $X$ to $Y$ consists of undirected edges and directed edges oriented towards $Y$.

[4]To see this, consider any DAG $G$ satisfying $Z \nrightarrow Y$. We can construct another DAG $G'$ from $G$ by adding the edge $Y \rightarrow Z$, since adding such an edge does not introduce a directed cycle. As a result, every topological ordering of $G'$ satisfies $Y < Z$, and $\mathcal{G}(Z \nrightarrow Y) \subseteq \mathcal{G}(Y < Z)$.

[5] The networks used in our experiments are available at `http://www.bnlearn.com/bnrepository`

[6] We select random sets of nodes and all their ancestors, up to a connected sub-network of a given size.

[7] Available at `http://www.maxsat.udl.cat/14/solvers/eva500a__`

[8] Available at `http://www.cs.york.ac.uk/aig/sw/gobnilp`

[9]When no limits are placed on the sizes of families (as was done here), heuristic-search approaches (like ours) have been observed to scale better than ILP approaches [Yuan and Malone, 2013, Malone et al., 2014].

[10]$\Delta$ can be greater than 0 when $p = 1$, as there may be many DAGs that respect a set of ancestral constraints. For example, DAG $X \rightarrow Y \rightarrow Z$ expresses the same ancestral relations, after adding edge $X \rightarrow Z$.

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
