[Reviews · NeurIPS 2016]

Reviewer 1

Summary

This paper concerns the constrained optimisation problem of learning Bayesian networks from local scores with ancestral constraints. The search is branch-and-bound where the nodes in the search tree are CPDAGs and we 'move down' from CPDAG1 to CPDAG2 by (roughly) choosing a parent set for a BN variable which is not a node in CPDAG1. Sensible symmetry breaking is effected. The search is A* where the URLearning system is used to provide a heuristic. (Details of how this happens are not given here but are available in an earlier paper).

Qualitative Assessment

Given ancestral constraints, some pruning of the search tree is possible. Lemma 3 (supplementary material) is the key result here. I believe it to be true, but I don't understand the proof. The phrase "By the EC tree edge generation rules, G_k also contains edge Z - > W" needs more explanation. In addition there are implied constraints ( "implied constraints" is the standard terminology, here they are called "projected constraints"). A bunch of ancestral constraints will imply constraints on edges which can be used for pruning. Constraints on permissible topological orderings are also inferred ("ordering constraints"). The approach to doing this is not properly explained. We have the statement that we "can infer Y < Z from Z not an ancestor of Y". The issue is the word "can". Clearly there are DAGs with Z is not an ancestor of Y but Y < Z is a consistent order: the graph with vertices Z and Y and no edges, for example. The authors, of course, know this! What they mean to say is that if "Z not an ancestor of Y" then we can *choose* to add the constraint Y < Z. So the idea is to 'infer' as many consistent ordering constraints as possible (not just those which are entailed). The authors are careful not to 'infer' too many and use a MAXSAT approach to do the inference. Experiments are conducted where the number of variables, datapoints and ancestral constraints are varied. There is a comparison with the GOBNILP system where the current system is shown to be far faster than GOBNILP. As the authors note, without ancestral constraints, GOBNILP performs worse than A* approaches when there is no limit on parent set cardinality. In the current experiments there is no such limit, so it is difficult to tell whether GOBNILP's poor performance here is due to its 'normal' failure to deal with many parent sets or is more specifically down to dealing with ancestral constraints poorly. Also the encoding of ancestral constraints ( given in the supplementary material) is based on a MAXSAT encoding from [Cussens, 2008] rather than, say, the ILP encoding found in Section 3.1.1 of [Cussens, 2010]. And this MAXSAT encoding is a poor formulation for an ILP system. Why not just have the A(X,Y) binary variables in addition to the I(Z,U) variables and omit the (cubic number of!) E(X,Y,Z) variables? I think all that is then needed is the transitivity constraints on the A(X,Y) variables and the following constraints linking A and I variables: A(X,Y) \geq \sum_{U:X \in U} I(Y,U) In other words if X is a parent of Y then X is an ancestor of Y. This formulation provides a far tighter linear relaxation and should provide much better performance. This is the approach of Section 3.1.1 of [Cussens, 2010] for a total ordering on variables but with a constraint dropped since ancestor is a partial, not total oder. I am confident that even with a sensible ILP formulation of ancestral constraints the current system will outperform GOBNILP, but the paper is weakened by comparing to a poor formulation rather than a sensible one. It would be interesting to compare the current method to that of van Beek and Hoffmann, particularly since it often outperforms GOBNILP. I don't think it would be too hard to adapt that method to allow ancestral constraints to be posted. The paper by van Beek and Hoffmann is not mentioned by the current authors. [Cussens, 2010] James Cussens. Maximum likelihood pedigree reconstruction using integer programming". WCB'10. Peter van Beek and Hella-Franziska Hoffmann. Machine learning of Bayesian networks using constraint programming. Proceedings of CP 2015, Cork, Ireland, August, 2015.

Confidence in this Review

3-Expert (read the paper in detail, know the area, quite certain of my opinion)


Reviewer 2

Summary

The authors present a method that leverages a given set of ancestral constraints to inform the (admissible) heuristic for an exhaustive search algorithm.

Qualitative Assessment

I found the experiments a bit confusing. Given the existing approaches of EC and GOB, Table 1 compares (EC + new work) to (GOB + new work'). If I understand this correctly, Table 3 is then comparing EC to GOB (or did I get this wrong?). It seems that it would make sense to do an explicit comparison between (EC + new work) and EC - although the (GOB + new work') vs GOB is interesting, it seems that EC is so much better than GOB that even if GOB benefited much more than EC from the new work, we would still want to go with (EC + new work).

Confidence in this Review

2-Confident (read it all; understood it all reasonably well)


Reviewer 3

Summary

The paper describes an efficient means to incorporate ancestral constraints in finding optimal Bayesian networks from data. It does this by translating a set of ancestral constraints into a (maximal) set of ordering constraints that are subsequently used in a score-based learning algorithm based on the EC tree introduced by [Chen et al., 2015]. Results are evaluated against a straightforward modification of the ILP based Gobnilp algorithm and shown to outperform this significantly in the given setting. Overall reasonably well written and clearly explained, although the solution is indeed conceptually straightforward. Novel problem + solution, nice result, not spectacular, but definitely of interest to a specialist audience.

Qualitative Assessment

The problem tackled in this paper is interesting and new in the sense that optimal BN learning in combination with ancestral constraints has no readily available ’standard solution’ (novelty=3). However, part of the reason behind this apparent deficiency is that the situation is a somewhat artificial setting: once ancestral relations come into play it would make more sense to go for more general causal discovery algorithms that also allow for unobserved confounders. (hence impact=2) The approach taken is based on translating hard-to-use ancestral constraints, that do not fit well in decomposable-score-based algorithms, into sets of ordering constraints that are more easily expressed in terms of absence of edges, which do fit decomposable scores. The mapping is done through a MaxSAT solver which ensures that all implied ancestral constraints are made explicit. However, some of the rules in 257-261 seem strange or incomplete: for example the premiss in acyclicity rule 3 is symmetric ( (X < Y) and (Z < W) == (Z < W) and (X < Y)), but the conclusion is not (Y -/-> Z); also rule 4 seems to exclude ancestral constraints of the type ‘X and Y do not cause each other’. Furthermore, one might expect that in practice only a few ancestral constraints are given, and these are unlikely to significantly reduce the search space. The resulting comparison against Gobnilp with added linear constraints looks impressive, but the latter is not developed to tackle this problem. A reasonable alternative to measure performance would be to assume that the ancestral constraints are in line with the optimal model, and compare against a greedy algorithm like GES which for oracle input should produce the same output. Of course greedy algorithms in general cannot guarantee optimality, but it would provide at least a more meaningful indication of the efficiency achieved by the method in the paper. Minor details: - 30: is there a difference between a (standard) oracle and an omniscient oracle? - 38: ‘empower the oracle’ = ‘use in combination with the oracle’ ? - 90: mention here optimizing = minimizing - 164: Theorem 1 should state more clearly that the indicated set of constraints A satisfied by CPDAG P on the EC tree is only a subset of the set of constraints A satisfied by the full DAG G (otherwise the iff does not apply) - 225-227: Theorem 3 seems unlikely to lead to many positive edge implications given sparse (small sets of) ancestral constraints. - 267: not easily recognisable as capturing ‘acyclic’ (see also above) - 310: for reasonable size graphs 10%-25% ancestral constraints would already imply extensive knowledge of the target graph (so not typical in practice)

Confidence in this Review

2-Confident (read it all; understood it all reasonably well)


Reviewer 4

Summary

This paper proposed a way to project the non-decomposable ancestral constraints to decomposable constraints so the decomposable-score-based structure learning approaches can incorporate such domain knowledge and hence improve the learning efficiency. The novelty of this paper mostly lies in section 5 where the authors introduced their proposed way to project the non-decomposable ancestral constraints to edge and topological ordering constraints which are decomposable.

Qualitative Assessment

The paper is well written and clearly illustrates their work both in the theory part and the experiments part. I have some concerns and suggestions as below: i. Table 3 shows that the projection of ancestral constraints can only improve the performance of EC trees but not the ILP approach. Does it mean that projecting the ancestral constraints to decomposable constraints only works for EC trees and have poor generalization ability to other learning approaches? Considering the main innovation of this paper is to infer decomposable constraints from non-decomposable ancestral constraints, it would be more convincing if the authors can show that incorporating such decomposable constraints can improve the performance and efficiency of certain learning approach which originally cannot handle the non-decomposable constraints. ii. Table 2 shows an easy-hard-easy pattern on the EC tree performance as increasing the proportion of ancestral constraints. Why does it happen? Does it mean that the effect of reducing the EC tree search space with ancestral constraints is fairly sensitive to the amount of domain knowledge it incorporates compared to the ILP approach? Hence, the number of ancestral constraints needs to be carefully tuned in order to achieve ideal performance? iii. Table 2 only shows the results of the structural Hamming distance from the EC trees. But, how does it compared with ILP approach? iv. A small suggestion: I understand that the BN graph contains the theoretical fundamental of EC trees as EC trees is extended from it. However, since the experiments part only considered the search space of EC trees with ancestral constraints applied to it, I would drop section 4.1 and spend more time on better explaining EC trees and the A* algorithm in the EC tree settings.

Confidence in this Review

2-Confident (read it all; understood it all reasonably well)


Reviewer 5

Summary

This paper considers the problem of finding an optimal Bayesian network configuration. In particular the authors show how to incorporate non-decomposable constraints (and in particular ancestral constrains) into the structure learning approach of Chen et al. [2015, 2016]. Empirical evaluation of the proposed approach against ILP based approaches shows orders of magnitude improvements.

Qualitative Assessment

The authors rephrase the problem as a MaxSAT problem with specific constraints. While my expertise in with this approach is limited the results (runtime comparison) seem quite impressive. Specifically the authors show a runtime reduction by orders of magnitude relative to alternative methods.

Confidence in this Review

1-Less confident (might not have understood significant parts)